# MALDI-TOF MS-Based Approaches for Direct Identification of Gram-Negative Bacteria and *Bla*_KPC_-Carrying Plasmid Detection from Blood Cultures: A Three-Year Single-Centre Study and Proposal of a Diagnostic Algorithm

**DOI:** 10.3390/microorganisms11010091

**Published:** 2022-12-29

**Authors:** Gabriele Bianco, Sara Comini, Matteo Boattini, Guido Ricciardelli, Luisa Guarrasi, Rossana Cavallo, Cristina Costa

**Affiliations:** 1Microbiology and Virology Unit, University Hospital Città della Salute e della Scienza di Torino, 10126 Turin, Italy; 2Department of Public Health and Paediatrics, University of Turin, 10124 Turin, Italy

**Keywords:** MALDI-TOF MS, Rapid Sepsityper, short-term subculture, blood culture, rapid testing, KPC variants, pKpQIL plasmid, subtyping, NG-Test^®^ CARBA 5, diagnostic algorithm

## Abstract

The rapid identification of pathogens of bloodstream infections (BSIs) and the detection of antibiotic resistance markers are critically important for optimizing antibiotic therapy and infection control. The purpose of this study was to evaluate two approaches based on MALDI-TOF MS technology for direct identification of Gram-negative bacteria and automatic detection of *Klebsiella pneumoniae carbapenemase* (KPC) producers using the Bruker MBT Subtyping IVD Module in a large routine laboratory over a three-year period. MALDI-TOF MS analysis was performed directly from blood culture (BC) bottles following bacterial pellet recovery by Rapid MBT Sepsityper^®^ Kit and on blood agar 4-h subcultures. Automated detection of *bla*_KPC_-carrying pKpQIL-plasmid by Bruker MBT Subtyping Module was evaluated in BCs tested positive to *K. pneumoniae* or *E. coli*. The results were compared with those obtained with conventional reference methods. Among the 2858 (93.4%) monomicrobial BCs, the overall species identification rates of the Rapid Sepsityper and the short-term subculture protocols were 84.5% (*n* = 2416) and 90.8% (*n* = 2595), respectively (*p* < 0.01). Excellent specificity for KPC-producers identification were observed for both MALDI-TOF MS protocols. The pKpQIL plasmid-related peak was detected in overall 91 of the 120 (75.8%) KPC-producing isolates. Notably, 14 out of the 17 (82.3%) *K. pneumoniae* isolates carrying *bla*_KPC_ variants associated with ceftazidime/avibactam resistance and tested negative by the immunocromatography assay, were correctly identified as KPC-producers by MALDI-TOF MS. In conclusion, combination of both Rapid Sepsityper and short-term subculture protocols may represent an optimal solution to promptly identify more than 95% of Gram-negative bacteria causing BSIs. MALDI Biotyper^®^ platform enabled a reliable and robust automated detection of KPC producers in parallel with species identification. However, integration of molecular or immunocromatographic assays are recommended according to local epidemiology.

## 1. Introduction

Bloodstream infections (BSIs) are serious medical conditions associated with high morbidity and mortality, for which rapid microbiological diagnostics is of utmost importance given the negative impact of delayed or inappropriate anti-infective treatment [1,2]. Although blood culture (BC) remains the gold standard method for BSI diagnosis, its main limitation is the long turnaround time. Several genotypic and phenotypic rapid testing performed directly from positive BC have been developed and introduced to the market over the last decade. These methods have not replaced conventional diagnostics but represent a complementary tool to promptly provide data regarding identification and antimicrobial susceptibility of BSI causative agents. Providing rapid microbiological results allows advance in effective antimicrobial regimen administration, which is related to a better outcome, especially for patients with severe sepsis or septic shock [3,4]. Moreover, simplifications and de-escalation of antimicrobial therapy are possible basing on rapid microbiological results, reducing unnecessary exposure to broad-spectrum antibiotics and risk of selecting drug-resistant microorganisms [4,5]. Several molecular systems have become available for the identification of microorganisms from BCs and/or detection of resistance markers [4,6,7,8,9]. DNA-microarray and Multiplex PCR-based assays, such as FilmArray^®^ system (BioFire Diagnostics, BioMérieux, Salt Lake City, UT, USA) and Verigene^®^ (Luminex^®^ Corporation, Chicago, IL, USA), respectively, allow simultaneous detection of many pathogens and antimicrobial resistance genes (e.g., *bla* carbapenemase genes, *bla*_CTX-M_, *mecA*, *vanA* and *vanB*) [6,7]. Although these systems can give results in 1–2.5 h and are easy to perform, their cost represents the main limitation to a routine use in BC diagnostics. Alternative diagnostic workflows include use of MALDI-TOF MS technology for microbial identification and rapid diagnostic assays for detection of antimicrobial resistance markers (e.g., molecular testing and lateral flow immunoassays) [10]. Over the past few years, MALDI-TOF MS has been widely employed as pathogen identification method. The instrument is available in most laboratories, and its use is well established in the everyday routine. Two main MALDI-TOF-based approaches were employed to obtain rapid microorganism identification from positive BC: direct identification from BC broth and identification using subculture on solid medium following short-term incubation [11]. In the first approach, as microorganisms in BC broth are not readily available for identification, an earlier step of extraction remains mandatory. Many extraction protocols are in-house protocols using several extraction steps and lysis reagents. However, in-house protocols may have a lack of standardization, often using modified cut-offs not validated by the MALDI-TOF-MS manufacture and requiring many steps with a time to results up to 40 min. The Sepsityper^®^ kit (Bruker Daltonik, Bremen, Germany) is a CE-IVD commercial assay to purify bacterial pellet from positive BC. The method involves the lysis of blood cells, followed by centrifugation and washing. According to a review and meta-analysis on its performance, it allows 79.8% of bacterial identification to the species level (76.1% and 89.6% for monomicrobial Gram-positive and Gram-negative BCs, respectively) in 30 min [12]. However, its important technical turnaround time, due to at least 5 centrifugation steps, limits its integration in routine procedures. A new version of Sepsityper^®^ kit named Rapid Sepsityper^®^ kit was introduced to the market to shorten hands-on time and identification delay to 10 min [10].

In addition to microorganism species identification, MALDI-TOF MS methodology was also used to identify antimicrobial resistance in bacterial pathogens through different approaches: 1. identification of antimicrobial-resistant clonal group (e.g., cfiA-positive *Bacteroides fragilis*, methicillin-resistant *S. aureus* and *vanA* gene-carrying *Enterococcus faecium* clones); 2. identification of a modified antimicrobial drug (e.g., carbapenemase and extended spectrum β-lactamases activity detection); 3. identification of the modified antimicrobial target (e.g., lipid A modification leading to colistin resistance in *E. coli* and *Acinetobacter* spp.); 4. direct detection of the antimicrobial resistance determinant (e.g., CMY-2 AmpC, and KPC-2 β-lactamases); 5. detection of biomarkers co-expressed with antibiotic resistant determinants (e.g., PSM-mec carrying MRSA and *bla*_KPC_-carrying plasmid) [13].

The aim of this study was to evaluate the performance of MALDI-TOF MS for direct identification of Gram-negative bacteria and KPC-producers directly from positive BCs to propose a rapid diagnostic algorithm for BC routine in a geographic area with endemic distribution of KPC-producing *K. pneumoniae*.

## 2. Materials and Methods

### 2.1. Study Design

This study was performed in a tertiary teaching hospital in Northwestern Italy over a three-year period (November 2019–October 2022). The BactAlert Virtuo instrument (BioMérieux, Marcy l’Ètoile, France) was used for BC processing during the study period. BCs tested positive to Gram-negative rods at microscope examination were enrolled for evaluation of MALDI-TOF MS technology as follows:(1) Microbial identification by MALDI-TOF MS was evaluated from both the BC bottle and the 4-hour blood agar subculture; (2) performance of MALDI Biotyper^®^ (Bruker Daltonik, Bremen, Germany) to identify KPC producers in *K. pneumoniae* and *E. coli* was evaluated.

Only one BC bottle per patient/BSI event was included in the analysis and samples collected from patients with previous Gram-negative BSI within the previous 10 days were excluded. BCs that tested polymicrobial were also excluded from the analysis. The results were compared with those obtained with conventional reference methods.

### 2.2. Microbial Identification by MALDI-TOF MS

Direct MALDI-TOF MS analysis was performed directly from BC bottles following bacterial pellet recovery by Rapid MBT Sepsityper^®^ IVD Kit. The manufacturer’s recommended procedure, with some modifications, was carried out as follows [10]: 1. transfer nominal 1 mL BC fluid to a 2 mL Eppendorf tube; 2. add nominal 200 μL Lysis Buffer and mix by vortexing for 10 (±5) seconds; 3. centrifuge the tube for 2 min at 13,000 rpm at room temperature; 4. remove the supernatant by pipetting and discard; 5. add nominal 1 mL Washing Buffer and resuspend the pellet by pipetting up and down; 6. centrifuge the tube for 1 min at 13,000 rpm at room temperature; 7. remove the supernatant from the pellet by pipetting and discard. 8. if the bacterial pellet is contaminated with material from the blood, add 1 mL of Washing Buffer, vortex for 10 (±5) seconds to suspend the bacterial pellet, allow the corpusculated material to settle for 30 s, transfer the supernatant suspension into a new 2 mL Eppendorf tube and proceed to step 6.

MALDI-TOF MS analysis on short-term subculture was performed on bacterial growth obtained on blood agar plate after 4-h incubation at 37 °C in 5% CO_2_ atmosphere.

For both methods, a small amount of bacteria (spindown pellet or short-term subculture) was transferred onto the MALDI-TOF steel target plate using a toothpick, coated with 1 μL of HCCA matrix, allowed to dry at room temperature and subjected to MALDI-TOF MS analysis on Bruker Microflex LT mass spectrometer. FlexControl 3.3 and Maldi Biotyper 3.0 software (Bruker Daltonik, Bremen, Germany) were applied to acquire the spectra and for the identification of the isolates with the Biotyper module, respectively. The Bacterial Test Standard (Bruker Daltonics) was used for calibration purposes. Each sample was tested in duplicate and species identification with highest confidence score from each pair was recorded. In cases of discordant results, the identification was considered unreliable. Confidence score values threshold ≥1.8 and <2, and ≥2 was accepted for identification to genera and species level, respectively.

MALDI-TOF MS analysis on pure overnight subcultures was considered as reference identification method.

### 2.3. Detection of KPC-Producing Klebsiella Pneumoniae or Escherichia coli

The potential of MALDI-TOF MS reaches beyond species identification and the MBT Subtyping IVD Module is the first IVD application exploring this potential. Herein, automated detection of KPC by MALDI-TOF MS was performed using MBT Subtyping IVD Module, which combines the identification of pathogens and subsequent detection of peak at 11,109 *m*/*z* in one automated workflow. This specific peak is related to *bla*_KPC_-carrying pKpQIL plasmid [14].

The IVD module of MBT subtyping has been implemented in our BC diagnostic workflow since January 2022, so automatic KPC detection was evaluated on all *E. coli* or *K. pneumoniae* positive BCs starting from this date. Prerequisite for the automated detection process with the IVD MALDI Biotyper was the reliable *E. coli* and *K. pneumoniae* identification with confidence identification score ≥2.0.

Performance of MALDI-TOF MS for KPC detection was evaluated comparing results with those obtained by reference molecular (Eazyplex^®^ SuperBug CRE; Amplex Diagnostics GmbH, Bremen, Germany) and immunochromatographic (NG-Test^®^ CARBA 5; NG Biotech, Guipry, France) assays [10].

Additionally, evaluation of KPC detection by MALDI-TOF MS was extended to spiked BCs. Overall, 200 consecutive *K. pneumoniae* isolates (KPC producers, *n* = 100, and ESBL producers, *n* = 100) from BC samples in the period 2020–2021 were used. Clinical BC bottles (BACT/ALERT^®^ FA/FN Plus bottles, BioMérieux, Marcy l’ Etoile, France) that remained negative after 5 days of incubation were anonymized and spiked as previously described [15]. For each isolate, both aerobic and anaerobic BC bottle were spiked. Next, spiked bottles were incubated in the automated system BacT/Alert Virtuo (BioMérieux) until the flask was flagged as positive. Flagged positive BCs were subjected to the MALDI-TOF MS analysis: 1. on bacterial pellet recovered from BC bottle by Rapid MBT Sepsityper^®^ IVD Kit; 2. on 4-h blood agar subculture; 3. on overnight subculture grown on MacConkey agar.

### 2.4. Statistical Analysis

Descriptive data were presented as absolute (*n*) and relative (%) frequencies. Sensitivity and specificity with 95% confidence interval (95% CI) were computed using MedCalc software version 16.8.4. Comparison involving dichotomous variables was tested using X2 test. Statistical significance was set a *p*-value < 0.05.

## 3. Results

### 3.1. Performance of Rapid Sepsityper and 4-h Short Subculture MALDI-TOF MS Protocols

In total, 3060 BCs tested positive to Gram-negative bacteria at microscopic examination and were subjected to microbial identification by both MALDI-TOF MS approaches. Of these, 202 (6.6%) were polymicrobial by examination of overnight subcultures and therefore were excluded from the analysis. Among the 2858 (93.4%) monomicrobial BCs, Enterobacterales were by far the most prevalent (78%, *n* = 2229) followed by non-fermenting Gram-negative species (20.1%; *n* = 525), obligate anaerobic species (1.9%; *n* = 54) and other fastidious species (1.7%; *n* = 50) (Table 1).

The overall species identification rates of the Rapid Sepsityper and the short-term subculture methods were 84.5% (*n* = 2416) and 90.8% (*n* = 2595), respectively (*p* < 0.01). Rates of species/genus misidentification were very low using both protocols, being 0.6% (*n* = 17) and 0.3% (*n* = 9), respectively (*p* = 0.11).

Among Enterobacterales, 89.4% (*n* = 1993) and 97.3% (*n* = 2169) were identified to the species and genus level by the Rapid Sepsityper method, while 96% (*n* = 2140) and 99% (*n* = 2206) performing analysis on short-term subcultures (*p* < 0.01), respectively. Considering the most representative species (frequency > 10), species identification rates ranged from 81.4% to 93.3% and from 82.3% to 100% for Rapid Sepsityper and short-term subculture protocols, respectively.

In the group of non-fermenters, the species and genus identification rates were, respectively, 66.5% (*n* = 349) and 77.1% (*n* = 405) by Rapid Sepsityper protocol and reached 82.3% (*n* = 432) and 92.8% (*n* = 487) using short-term subcultures (*p* < 0.01), respectively. The species identification rates varied substantially between species. The best performance was observed for *Pseudomonas aeruginosa* (78.7% and 93.9% using Rapid Sepsityper and short-term subculture protocols, respectively, *p* < 0.01).

Rapid Sepsityper protocol correctly identify 29.6% and 42.6% of obligate anaerobic species at species and genus level, respectively. Conversely, short-term subculture protocol was not performed since no growth was obtained on agar blood subcultures incubated at 37 °C in 5% CO_2_ atmosphere. For fastidious species (mostly *Campylobacter* spp. and *Moraxella osloensis*), the identification rate at species level was 26% and 46% by rapid Sepsityper and short-term subculture protocols, respectively (*p* < 0.01).

### 3.2. Performance of MALDI-TOF MS in Detection of KPC Producers

Considering the total of Enterobacterales isolates, the rate of carbapenemase carriage was 8.9% (198/2229) according to reference conventional methods. The main carbapenemases detected were KPC (89.9%; *n* = 178), followed by Verona integron-encoded metallo-beta-lactamase (VIM) (50.5%; *n* = 10), new New Delhi Metallo-beta-lactamase (NDM) (2%; *n* = 4), Oxacillinase-48-like (OXA-48-like) (1.5%; *n* = 3) and KPC + VIM (1.5%, *n* = 3). No Imipenemase-type (IMP-type) carbapenemase was detected. *K. pneumoniae* was the main species carrying carbapenemase genes (185/198, 93.4%) (*p* < 0.01). Overall, 99.4% (177/178) of *bla*_KPC_ were detected in *K. pneumoniae* isolates. Of note, 17 out of the 177 (9.6%) *K. pneumoniae* isolates carrying *bla*_KPC_ (9.5%) tested negative by NG-Test^®^ CARBA 5 immunochromatographic assay.

Performances of MALDI-TOF MS in detecting KPC producers among *K. pneumoniae* and *E. coli* using Rapid Sepsityper and short-term subculture protocols are presented in Table 2. No significant discrepancy was observed between clinical and spiked BCs and according to the type of bottle used (BACT/ALERT^®^ FA and FN Plus). Isolates tested negative to KPC carriage included producers of carbapenemase other than KPC (*K. pneumoniae*: NDM, *n* = 2; VIM, *n* = 5; OXA-48-like, *n* = 1; *E. coli*: NDM, *n* = 2; VIM, *n* = 3; OXA-48-like, *n* = 2) and extended spectrum β-lactamases (*K. pneumoniae*: *n* = 120; *E. coli*: *n* = 62).

The pKpQIL plasmid-related peak was detected by Biotyper software in overall 91 of the 120 KPC-producing isolates (75.8%). No difference in sensitivity rate between Rapid Sepsityper and short-term subculture protocols was observed, and results were confirmed by performing the test on overnight subcultures. Specificity was 99.8% and 100% for Rapid Sepsityper and short-term subculture protocols, respectively. Notably, 14 out of the 17 (82.3%) *K. pneumoniae* isolates carrying *bla*_KPC_ which tested negative by immunochromatographic assay, were correctly identified as KPC-producers by MALDI-TOF MS. One false positive result was obtained by Rapid Sepsityper protocol in a clinical BC; this result was not confirmed by MALDI-TOF MS analysis performed on overnight subculture.

## 4. Discussion

BC rapid diagnostics is an essential tool for timely optimization of antimicrobial management of BSIs patients, especially now that new antimicrobial molecules (e.g., ceftazidime/avibactam, meropenem/vaboarbactam, imipenem/relebactam and cefiderocol) are available for the treatment of multidrug-resistant bacteria infections. The implementation of fast, easy, and cost-effective diagnostic tests is a feasible solution for most microbiology laboratories. Although many molecular testing approaches are easy to perform and have a very short turnaround time, they are not cheap and their use is mainly reserved to critically ill patients [8,9]. Furthermore, the number of detectable microbial targets by commercial multiplex assays (e.g., BioFire FilmArray Blood Culture Identification (BCID) panel-2 (BCID-2) (BioFire Diagnostics) is limited [6,7]. In this study, we evaluated two MALDI TOF MS approaches for the rapid identification of Gram-negative organisms from BCs, and the application of automatic MALDI-TOF MS detection of pKpQIL plasmid-related peak for identification of *E. coli* and *K. pneumoniae* KPC-producers isolates.

Rapid Sepsityper Kit (Bruker Daltonics GmbH, Bremen, Germany) has been designed to shorten the processing time without affecting diagnostic performance. Herein, Rapid Sepsityper protocol accurately identified 84.5% and 91.7% of the 2858 blood culture Gram-negative isolates at species and genus level, respectively. This trend with Sepsityper Kit or Rapid Sepsityper Kit has been reported in previous studies [10,12,16,17,18]. However, performance varied according to bacterial species. As previously reported [10], matched identification rate between the conventional and Rapid Sepsityper protocol was significantly higher for Enterobacterales (97.3%) in comparison to non-fermenting Gram-negative species (77.1%). Rapid Sepsityper protocol displayed poor identification performance for samples positive to anaerobe obligate and fastidious species group (42.6% and 48%, respectively). A previous study using the Sepsityper method has shown lower species identification rate of anaerobes than saponin method (56.3% vs. 84.9%) [19]. D’Inzeo et al. reported 82.4% of correct identification (log (score) values ≥ 1.9) of 170 Gram-negative anaerobe obligate BC isolates (mostly *Bacteroides* spp.) by MALDI TOF MS analysis on bacterial pellet recovered from 8 mL of BC fluid by a non-saponin-based extraction protocol [20]. Thus, performance of direct identification of anaerobes could be influenced by the sample preparation method used, and protocols involving a larger volume of BC broth could be suggested to achieve better performance.

MALDI-TOF MS analysis performed on blood agar subculture following 4-h incubation showed a better performance than Rapid Sepsityper protocol. Correct genus/species identification rate by the short-term subculture protocol was 99%, 92.8% and 64%, respectively, for Enterobacterales, non-fermenting species and fastidious species group, significantly higher than those obtained with the Rapid Sepsityper protocol (97.3%, 77.1% and 24.5%, respectively).

In choosing the most appropriate MALDI-TOF MS protocol for rapid identification in BC routine, some considerations should be made. Despite both protocols can provide results within the same day of BC positivity detection, Rapid Sepsityper Kit has a reduced turnaround time (20 min) but a lower performance than short-term subculture protocol (~4 h). This latter does not require sample preparation steps resulting in reducing the hands-on processing time and the cost of identification but is not applicable for obligate anaerobes if no subculture in anaerobic atmosphere has been set up.

The possibility of using the MALDI-TOF MS mass spectrum to identify specific determinants of resistance in addition to the bacterial species represents a cost-effective method to timely obtain useful data both for therapeutic and infection control purposes. In this study, ability of the MBT Subtyping IVD module to identify KPC-producers through detection of pKpQIL plasmid related-peak in an area with high endemicity of KPC-producing *K. pneumoniae* was assessed [21,22]. The sensitivity was found to be 75.8% and was not affected by the MALDI-TOF MS protocols used. This finding indicates that KPC-producing harboring pKpQIL plasmid represents the main circulating clones in our hospital. The method showed high specificity, classifying all non-KPC-carbapenamase-, ESBL- and non-ESBL/carbapenamse-producers as KPC-peak negative. Interestingly, the method was able to identify 14 out of the 17 KPC-producing *K. pneumoniae* isolates that tested negative to NG-Test^®^ CARBA 5 immunoassay but positive to Eazyplex^®^ SuperBug CRE molecular testing. Nine of these isolates were previously investigated by DNA sequencing revealing mutations in *bla*_KPC_ associated with ceftazidime/avibactam resistance (substitution D179Y, *n* = 8; duplication EL167–168, *n* = 1) [23,24,25]. KPC variants conferring ceftazidime/avibactam resistance are an increasing concern as characterized to be undetectable by the main phenotypic carbapenemase detection methods including immunochromatographic assays [23,24,26]. Since phenotypic assays represent the most used and cost-saving method to identify carbapenemase producers in clinical microbiology laboratories, failure to recognize these mutated-KPC-producing strains could facilitate their spread in hospital setting [22,27]. Hence, automated MALDI-TOF MS KPC detection may represent an alternative method to molecular testing to detect KPC variants when expressed by *bla*_KPC_ carried by pKpQIL plasmid.

Since *bla*_KPC_ can be present in divergent plasmids and sometimes in chromosomes, *bla*_KPC_-carrying plasmid-associated biomarker cannot be used as an exclusive method for KPC detection and should be integrated with additional phenotypic and/or genotypic methods [28]. In addition, it should be considered that the trust identification score value ≥ 2.0 is a prerequisite for the automatic detection process with MALDI Biotyper and that the method has “in vitro diagnostic” (IVD) validation at present only for *K. pneumoniae* and *E. coli*.

Considering the results obtained from this study, we proposed a rapid diagnostic algorithm for Gram-negative bacteria positive BCs primarily based on MALDI-TOF MS technology (Figure 1). Given the rapid execution and short turnaround time, MALDI-TOF MS analyses might be promptly performed using Rapid Sepsityper protocol directly from BC positive to Gram-negative bacteria by microscopic examination. In case of unreliable identification (score < 1.80) or reliable only at the genus level (score ranging from 1.80 to 1.99), short-term subculture MALDI-TOF MS protocol should be performed. With both protocols, in case of reliable identification (score > 2.00) of *K. pneumoniae* or *E. coli*, *bla*_KPC_-carrying plasmid-associated biomarker detection will be suggestive of KPC-producing strain identification. Conversely, if KPC-biomarker will not be detected, additional immunochromatographic and/or molecular testing should be performed to confirm KPC negativity. Molecular testing should be considered if KPC variants not detectable by immunochromatographic assay are suspected.

This study supports evidence for the real time detection of a resistance marker in parallel to species identification by using MALDI Biotyper platform. However, there are some limitations. A larger-scale test should be needed to obtain more accurate data. Although more than 2000 isolates were included in the study, the sample sizes of fastidious and anaerobe species were relatively small. Furthermore, numbers of *K. pneumoniae* and *E. coli* included for evaluation of MALDI Biotyper^®^ subtyping for KPC-producers detection were limited, and no KPC-producing *E. coli* were included. The rapid diagnostic BC algorithm proposed was designed according to our local epidemiology. Sensitivity of the subtyping approach depends on the regional epidemiology of KPC-producing strains including prevalence of pKpQIL plasmid-encoded KPC. Integration of methods to detect non-pKpQIL-encoded KPC or carbapenemases other than KPC is recommended according to local epidemiology. Further studies are warranted to evaluate the method to bacterial species other than *K. pneumoniae*.

## 5. Conclusions

The combination of Rapid Sepsityper and short-term subculture protocols may represent an optimal solution to promptly identify more than 95% of Gram-negative bacteria causing BSI on the same day that BC positivity is detected. Our study shows that MALDI Biotyper^®^ platform enabled a reliable and robust automated detection of KPC-producing *K. pneumoniae* isolates in parallel with species identification, without additional procedures or additional costs. This method showed being applicable to bacterial colony, short-term subculture as well as directly on BC bacterial pellet. The implementation of this MALDI-based approach is feasible in high throughput routine laboratories. However, the integration of molecular or immunocromatographic assays is recommended according to local diffusion of non-pKpQIL-encoded KPC and carbapenemases other than KPC.

## Figures and Tables

**Figure 1 microorganisms-11-00091-f001:**
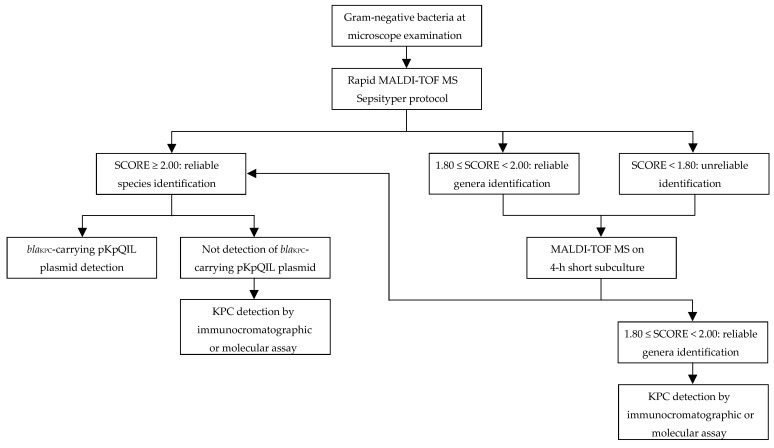
Blood cultures rapid diagnostic algorithm for identification of Gram-negative bacteria and KPC producers.

**Table 1 microorganisms-11-00091-t001:** MALDI-TOF MS identification of Gram-negative isolates from monomicrobial positive blood cultures using Rapid Sepsityper and 4-h short-subculture protocols.

Conventional Method (No. of Isolates)		Rapid Sepsityper (No. of Identified Isolates; %)			4 h-Short Subculture (No. of Identified Isolates; %)		
Gram-Negative Organism		Species Level (Score ≥ 2.0)	Genus Level (Score ≥ 1.8)	Mis-Identification; (%)	Species Level (Score ≥ 2.0)	Genus Level (Score ≥ 1.8)	Mis-Identification; (%)
Enterobacterales	2229	1993; 89.4%	2169, 97.3%	9; 0.4%	2140; 96%	2206; 99%	5; 0.2%
*Escherichia coli*	1031	961; 93.2%	1017; 98.6%	3; 0.3%	1009; 97.9%	1029;99.8%	0
*Klebsiella pneumoniae*	580	525; 90.5%	572; 98.6%	2; 0.3%	561; 96.7%	572; 98.6%	0
*Enterobacter cloacae* complex	183	149; 81.4%	176; 96.2%	1; 0.5%	169; 92.3%	178; 97.3%	1; 0.5%
*Serratia marcescens*	97	82; 84.5%	92; 94.8%	0	90; 92.8%	93; 95.9%	0
*Klebsiella oxytoca*	72	61; 84.7%	69; 95.8%	1; 1.4%	70; 97.2%	72; 100%	0
*Proteus mirabilis*	67	59; 88%	63; 94%	0	62; 92.5%	66; 98.5%	0
*Klebsiella aerogenes*	62	51; 82.2%	56; 90.3%	0	58; 93.5%	62; 100%	0
*Citrobacter koseri*	32	27; 84.4%	30; 93.7%	0	30; 93.7%	31; 96.9%	0
*Citrobacter freundii* complex	22	19; 86.4%	21; 95.4%	1; 4.5%	21; 95.4%	22; 100%	0
*Morganella morganii*	17	14; 82.3%	15; 88.2%	0	14; 82.3%	16; 94.1%	0
*Klebsiella variicola*	15	14; 93.3%	14; 93.3%	1; 6.7%	14; 93.3%	14; 93.3%	1; 6.7%
*Salmonella enterica*	9	7; 77.8%	8; 88.9%	0	9; 100%	9; 100%	0
*Pantoea agglomerans*	8	5; 62.5%	7; 87.5%	0	7; 87.5%	8; 100%	0
*Hafnia alvei*	8	6; 75%	8; 100%	0	6; 75%	8; 100%	0
*Proteus vulgaris*	6	4; 66.7%	5; 83.3%	0	6; 100%	6; 100%	0
*Pantoea septica*	3	2; 66.7%	2; 66.7%	0	2; 66.7%	3; 100%	0
*Serratia liquefaciens*	3	1; 33.3%	2; 66.7%	0	2; 66.7%	3; 100%	0
*Pantoea eucrina*	2	1; 50%	2; 100%	0	2; 100%	2; 100%	0
*Providencia stuartii*	2	1; 50%	2; 100%	0	2; 100%	2; 100%	0
*Raoultella ornithinolytica*	2	1; 50%	2; 100%	0	1; 50%	2; 100%	0
*Serratia rubidea*	2	1; 50%	1; 50%	0	2; 100%	2; 100%	0
*Enterobacter bugandensis*	2	1; 50%	2; 100%	0	1; 50%	2; 100%	0
*Providencia rettgeri*	1	0	1; 100%	0	1; 100%	1; 100%	0
*Yersinia enterocolitica*	1	1; 100%	1; 100%	0	1; 100%	1; 100%	0
*Proteus penneri*	1	0	0	0	0	1; 100%	0
*Klebsiella ozaenae*	1	0	1; 100%	0	0	1; 100%	0
Non-fermenting species	525	349; 66.5%	405; 77.1%	7; 1.3%	432; 82.3%	487; 92.8%	3; 0.6%
*Pseudomonas aeruginosa*	310	244; 78.7%	272; 87.7%	1; 0.3%	291; 93.9%	305; 98.4%	0
*Acinetobacter baumannii* complex	136	80; 58.8%	94; 69.1%	1; 0.7%	101; 74.3%	121; 89%	1; 0.7%
*Stenotrophomonas maltophilia*	22	11; 50%	15; 68.2%	0	17; 77.3%	19; 86.4%	0
*Achromobacter xylosoxidans*	9	5; 55.5%	7; 77.8%	1; 11.1%	6; 66.7%	8; 88.9%	1; 11.1%
*Acinetobacter lwoffii*	6	2; 33.3%	3; 50%	1; 16.7%	4; 66.7%	4; 66.7%	1; 16.7%
*Aeromonas caviae*	4	2; 50%	2; 50%	0	2; 50%	3; 75%	0
*Acinetobacter ursingii*	4	1; 25%	1; 25%	0	2; 50%	3; 75%	0
*Ochrobactrum anthropi*	3	1; 33.3%	1; 66.7%	0	1; 33.3%	1; 33.3%	0
*Paracoccus yeei*	3	0	0	1; 33.3%	1; 33.3%	2; 66.7%	0
*Burkholderia cenocepacia*	2	0	1; 50%	1; 50%	1; 50%	2; 100%	0
*Pseudomonas putida*	2	1; 50%	1; 50%	0	1; 50%	2; 100%	0
*Pseudomonas oryzihabitans*	2	0	1, 50%	0	0	1; 50%	0
*Rhizobium radiobacter*	2	0	0	0	0	1; 50%	0
*Acinetobacter radioresistens*	2	0	1; 50%	0	0	1; 50%	0
*Aeromonas veronii*	1	0	1; 100%	0	0	1; 100%	0
*Pseudochrobactrum asaccharolyticum*	1	0	0	1; 100%	0	0	0
*Pseudomonas chlororaphis*	1	0	0	0	0	1; 100%	0
*Pseudomonas pseudomallei*	1	0	0	0	0	1; 100%	0
*Pseudomonas monteilii*	1	0	0	0	0	1; 100%	0
*Pseudomonas kuykendallii*	1	0	0	0	0	1; 100%	0
*Pseudomonas luteola*	1	0	0	0	1; 100%	1; 100%	0
*Stenotrophomonas rhizophila*	1	0	1; 100%	0	1; 100%	1; 100%	0
*Acinetobacter vivianii*	1	0	0	0	0	1; 100%	0
*Acinetobacter guillouiae*	1	0	0	0	0	1; 100%	0
*Acinetobacter variabilis*	1	0	0	0	0	0	0
*Acinetobacter radioresistens*	2	1; 50%	1; 50%	0	1; 50%	1; 50%	0
*Acinetobacter lactucae*	2	1; 50%	1; 50%	0	1; 50%	1; 50%	0
*Roseomonas mucosa*	1	0	0	0	0	1; 100%	0
*Cupriavidus basilensis*	1	0	1; 100%	0	0	1; 100%	0
*Raoultella ornithinolityca*	1	0	1; 100%	0	1; 100%	1; 100%	0
Obligate anaerobic species	54	16; 29.6%	23; 42.6%	1; 1.8%	-	-	-
*Bacteroides fragilis*	37	12; 32.4%	16; 43.2%	1; 2.7%	-	-	-
*Bacteroides vulgatus*	3	1; 33.3%	2; 66.7%	0	-	-	-
*Leptotrichia spp.*	2	0	0	0	-	-	-
*Bacteroides thetaiotaomicron*	2	0	0	0	-	-	-
*Fusobacterium gonidiaformans*	2	1; 50%	1; 100%	0	-	-	-
*Parabacteroides goldsteinii*	1	1; 100%	1; 100%	0	-	-	-
*Prevotella denticola*	1	1; 100%	1; 100%	0	-	-	-
*Prevotella melaninogenica*	1	0	1; 100%	0	-	-	-
*Prevotella baroniae*	1	0	0	0	-	-	-
*Bacteroides uniformis*	1	0	0	0	-	-	-
*Bacteroides pyogenes*	1	0	0	0	-	-	-
*Bacteroides ovatus*	1	0	1; 100%	0	-	-	-
*Veillonella dispar*	1	0	0	0	-	-	-
Fastidious species	50	13; 26%	24; 48%	0	23; 46%	32; 64%	1; 2%
*Campylobacter jejuni*	19	5; 26.3%	7; 36.8%	0	9; 47.4%	12; 63.1%	1; 5.3%
*Moraxella osloensis*	10	4; 40%	7; 70%	0	8; 80%	9; 90%	0
*Haemophilus influenzae*	5	1; 20%	2; 40%	0	2	2; 40%	0
*Campylobacter fetus*	4	1; 25%	2; 50%	0	0	2; 50%	0
*Campylobacter coli*	2	1; 50%	2; 100%	0	1	2; 100%	0
*Leptotrichia trevisanii*	2	0	0	0	0	0	0
*Moraxella liquefaciens*	2	0	1; 50%	0	0	2; 100%	0
*Eikenella corrodens*	1	0	0	0	0	0	0
*Capnocytophaga canimorsus*	1	0	0	0	0	0	0
*Capnocytophaga sputigena*	1	0	1; 100%	0	1; 100%	1; 100%	0
*Neisseria meningitidis*	1	0	0	0	0	0	0
*Neisseria gonorrhoeae*	1	1; 100%	1; 100%	0	1; 100%	1; 100%	0
*Moraxella nonliquefaciens*	1	0	1; 100%	0	1; 100%	1; 100%	0
Total isolates	2858	2416, 84.5%	2621, 91.7%	17; 0.6%	2595; 90.8%	2725;95.3%	9; 0.3%

**Table 2 microorganisms-11-00091-t002:** The performance of Bruker MBT Subtyping IVD Module for rapid identification of KPC-producing *Klebsiella pneumoniae* or *Escherichia coli*.

		Rapid Sepsityper Protocol				4 h-Short Subculture Protocol			
KPC Phenotype/Genotype According to Reference Methods		Positive	Negative	Sensitivity [95% CI]	Specificity [95% CI]	Positive	Negative	Sensitivity [95% CI]	Specificity [95% CI]
Clinical BCs									
*K.pneumoniae* (*n* = 110)	KPC-positive	16	4	80% [56.3–94.3%]	98.9% [94–100%]	16	4	80% [56.3–94.3%]	100% [96–100%]
	KPC-negative	1	89			0	90		

*E. coli* (*n* = 230)	KPC-positive	0	0	-	100% [98.4–100%]	0	0	-	100% [98.4–100%]
	KPC-negative	0	230			0	230		

Spiked BCs									
*K.pneumoniae* (*n* = 200)	KPC-positive	75	25	75% [65.3–83.1%]	100% [96.4–100%]	75	25	75% [65.3–83.1%]	100% [96.4–100%]
	KPC-negative	0	100			0	100		
Total (*n* = 540)	KPC-positive	91	29	75.8% [67.2–83.2%]	99.8% [98.7–100%]	91	29	75.8% [67.2–83.2%]	100% [99.1–100%]
	KPC-negative	1	419			0	420		

Abbreviations: CI, confidence interval; BCs, blood cultures.

## Data Availability

The authors confirm that the data supporting the findings of this study are available from the corresponding author on reasonable request.

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
