# Peer review of "MALDI-TOF MS-Based Approaches for Direct Identification of Gram-Negative Bacteria and BlaKPC-Carrying Plasmid Detection from Blood Cultures: A Three-Year Single-Centre Study and Proposal of a Diagnostic Algorithm"

_microorganisms, 2022, doi:10.3390/microorganisms11010091_

Round 1
Reviewer 1 Report
The manuscript entitled "MALDI-TOF MS-based approaches for direct identification of Gram-negative bacteria and blaKPC-carrying plasmid detection from blood cultures: a three-year single-centre study and proposal of a diagnostic algorithm" by Bianco and co-authors describe the proposal of a new diagnostic methodology based on MALDI-TOF MS techniques to automatically identify Escherichia coli and Klebsiella pneumoniae blaKPC-carrying pKpQIL-plasmid. The manuscript is well organized and written, on a topic of relevance for clinical microbiology, as the fast identification of the ethiological agent responsible for blood infections is determinant for the patient survival.
There are however some limitations to the study. For instance, no carbapenemase-producing E.coli was included in the study performed using the Bruker MBT Subtyping IVD Module for rapid identification of KPC-producing Klebsiella pneumoniae or Escherichia coli. The missing of this control in my view limits any conclusion that can be taken concerning the utility of the methods for fast detection of carbapenemase-producing E. coli.
The presentation of data in Table 1 is confusing as the authors chose to present the number of isolates followed by "," and then the percentage. I suggest to use ";" as this would not allow confusions. The last raw of table 1 indicates "Total species" and I presume this is a mistake, total isolates?
In Table 2, in the first column, it is indicated positive and negative. This is also confusing as on the talbe first raw iit is also indicated positive and negative. A possibility to solve this confusion maybe to use carbapenemase-positive and carbapenemase -negative in the first column and save the"positive and negative for the assy results.
In the discussion section, the first sentence states: "BC rapid diagnostics is an essential tool for timely optimize antimicrobial management of BSIs patients , especially now that new antimicrobial molecules are available for the treatment of multidrug-resistant bacteria infections." The sentence should be reviewed as presently antimicrobials to treat multidrug resistant bacteria are scarce.
Minor issues:
There are several abbreviations used in the abstract without any explanation. The abbreviations need to be explained.
Throughout the manuscript several bacterial species names are not italicized. This needs to be corrected.
Author Response
Reviewer: 1
1.The manuscript entitled "MALDI-TOF MS-based approaches for direct identification of Gram-negative bacteria and blaKPC-carrying plasmid detection from blood cultures: a three-year single-centre study and proposal of a diagnostic algorithm" by Bianco and co-authors describe the proposal of a new diagnostic methodology based on MALDI-TOF MS techniques to automatically identify Escherichia coli and Klebsiella pneumoniae blaKPC-carrying pKpQIL-plasmid. The manuscript is well organized and written, on a topic of relevance for clinical microbiology, as the fast identification of the ethiological agent responsible for blood infections is determinant for the patient survival.
We thank the referee for these appraisals.
2. There are however some limitations to the study. For instance, no carbapenemase-producing E.coli was included in the study performed using the Bruker MBT Subtyping IVD Module for rapid identification of KPC-producing Klebsiella pneumoniae or Escherichia coli. The missing of this control in my view limits any conclusion that can be taken concerning the utility of the methods for fast detection of carbapenemase-producing E. coli.
We thank the referee for these comments. Limitations of the study are reported in Discussion (see lines 323-334). “However, there are some limitations. A larger-scale test should be needed to obtain more accurate data. Although more than 2,000 isolates were included in the study, the sample sizes of fastidious and anaerobe species were relatively small. Furthermore, numbers of K. pneumoniae and E.coli included for evaluation of MALDI Biotyper® subtyping for KPC-producers detection were limited, and no KPC-producing E. coli were included. The rapid diagnostic BC algorithm proposed was designed according to our local epidemiology. Sensitivity of the subtyping approach depends on the regional epidemiology of KPC-producing strains including prevalence of pKpQIL plasmid-encoded KPC. Integra-tion of methods to detect non-pKpQIL-encoded KPC or carbapenemases other than KPC is recommended according to local epidemiology. Further studies are warranted to evaluate the method to bacterial species other than K. pneumoniae.”
3. The presentation of data in Table 1 is confusing as the authors chose to present the number of isolates followed by "," and then the percentage. I suggest to use ";" as this would not allow confusions. The last raw of table 1 indicates "Total species" and I presume this is a mistake, total isolates?
We thank the referee for these comments. Accordingly, we revised Table 1.
4. In Table 2, in the first column, it is indicated positive and negative. This is also confusing as on the talbe first raw iit is also indicated positive and negative. A possibility to solve this confusion maybe to use carbapenemase-positive and carbapenemase -negative in the first column and save the"positive and negative for the assy results.
We thank the referee for this comment. Accordingly, we revised Table 2.
5. In the discussion section, the first sentence states: "BC rapid diagnostics is an essential tool for timely optimize antimicrobial management of BSIs patients , especially now that new antimicrobial molecules are available for the treatment of multidrug-resistant bacteria infections." The sentence should be reviewed as presently antimicrobials to treat multidrug resistant bacteria are scarce.
We thank the referee for this comment. New drugs are now available for treatment of MDR Gram-negative infections (ceftazidime/avibactam, meropenem/vaborbactam, imipenem/relebactam, cefiderocol). Rapid identification of mechanisms of resistance (e.g., type of carbapenemase) may be essential for prompt use of these new drugs, even before traditional antibiogram results.
6. Minor issues:
There are several abbreviations used in the abstract without any explanation. The abbreviations need to be explained.
Throughout the manuscript several bacterial species names are not italicized. This needs to be corrected.
We thank the referee for these comments. Accordingly, we revised the text.

Reviewer 2 Report
New tools for identifying bacteria are essential, especially in the era of resistance and post-antibiotics. The authors present an alternative and algorithm. The work is pertinent, however it should be revised.
1. Avoid one-sentence paragraphs. Expand the idea or connect with the previous paragraph.
2. Why is part of Table 2 in bold? What is the authors' aim?
3. The quality of figure 1 can be improved.
4. line 237. What are the new molecules?
5. The authors justify the need for new bacteria identification tools based on the cost of current protocols. What is the authors' view on the cost of a mass spectrometer? Most laboratories and hospitals in various parts of the world, such as Latin America, lack this equipment. Can the authors estimate the cost of this alternative and compare by presenting values?
6. Conclusion: "...cost-effective solution"... The data presented does not validate the cost.
7. Revise "easy-to-use"...Mass spectrometer operation requires training and qualification.
Author Response
Reviewer 2:
New tools for identifying bacteria are essential, especially in the era of resistance and post-antibiotics. The authors present an alternative and algorithm. The work is pertinent, however it should be revised.
We thank the referee for these appraisals.
1. Avoid one-sentence paragraphs. Expand the idea or connect with the previous paragraph.
We thank the referee for this comment. Accordingly, we revised the text
2. Why is part of Table 2 in bold? What is the authors' aim?
We thank the referee for this comment. It's a mistake. Accordingly, we revised the Tables.
3. The quality of figure 1 can be improved. Accordingly, we revised the Figure 1.
We thank the referee for this comment.
4. line 237. What are the new molecules?
We thank the referee for this comment. Accordingly we revised the sentence as follows: “BC rapid diagnostics is an essential tool for timely optimize antimicrobial manage-ment of BSIs patients , especially now that new antimicrobial molecules (e.g. ceftazidime/avibactam, meropenem/vaboarbactam, imipenem/relebactam and cefidero-col) are available for the treatment of multidrug-resistant bacteria infections.
5. The authors justify the need for new bacteria identification tools based on the cost of current protocols. What is the authors' view on the cost of a mass spectrometer? Most laboratories and hospitals in various parts of the world, such as Latin America, lack this equipment. Can the authors estimate the cost of this alternative and compare by presenting values?
We thank the referee for this comment. We revised the sentence as follows: “The combination of Rapid Sepsityper and short-term subculture protocols may rep-resent an optimal solution to promptly identify more than 95 percent of Gram-negative bacteria causing BSI on the same day that BC positivity is detected.” Assessment of instrumentation costs are beyond the scope of the study. MALDI-TOF MS is now widespread in most laboratories, and its costs excluding instrumentation are very limited. The aim of the study was to evaluate MALDI-TOF MS technology for species identification and KPC detection in blood cultures.
6. Conclusion: "...cost-effective solution"... The data presented does not validate the cost.
We thank the referee for this comment. We revised the sentence as follows: “The combination of Rapid Sepsityper and short-term subculture protocols may rep-resent an optimal solution to promptly identify more than 95 percent of Gram-negative bacteria causing BSI on the same day that BC positivity is detected.”
6. Revise "easy-to-use"...Mass spectrometer operation requires training and qualification.
We thank the referee for this comment. We revised the sentence as follows: “The combination of Rapid Sepsityper and short-term subculture protocols may rep-resent an optimal solution to promptly identify more than 95 percent of Gram-negative bacteria causing BSI on the same day that BC positivity is detected.”

Reviewer 3 Report
In this study, the authors employ a combination of Rapid Sepsityper kit and MALDI-TOF MS to identify bacterial cultures in BC. This is a timely study with acute importance for the identification of infectious agents in BC in hospitals and clinical laboratories. The manuscript is well-written and concise. A few minor corrections listed below should be addressed.
What was the rationale for excluding polymicrobial cultures? This should be mentioned in lines 109-110.
Line 195: the bacterial name should be in italics
Several acronyms are used throughout the manuscript. They should be completely written at least the first time of occurrence. Eg: klebsiella pneumoniae carbapenemase (KPC) and verona integron-encoded metallo-beta-lactamase (VIM)
Line 236: Should read as “timely optimization of….”
Line 240: Should read as “many molecular testing approaches…”
How does the sensitivity and specificity observed with MALDI-TOF (this study) compare with that of other established tests such as BCID?
Line 327: Italicize E. coli
The spelling of “Sepsityper” should be corrected in the figure.
Author Response
Reviewer 3:
In this study, the authors employ a combination of Rapid Sepsityper kit and MALDI-TOF MS to identify bacterial cultures in BC. This is a timely study with acute importance for the identification of infectious agents in BC in hospitals and clinical laboratories. The manuscript is well-written and concise. A few minor corrections listed below should be addressed.
We thank the referee for these appraisals.
1. What was the rationale for excluding polymicrobial cultures? This should be mentioned in lines 109-110.
We thank the referee for this comment. The purpose of the study was to evaluate the performance of MALDI technology on single-microbial blood cultures. Therefore, polymicrobial blood cultures were excluded from the analysis. This is a criterion reported in Materials and Methods and used in most studies that have evaluated the MALDI-TOF MS. Further studies will be performed for polymicrobial blood cultures whose frequency is small.
2. Line 195: the bacterial name should be in italics
Several acronyms are used throughout the manuscript. They should be completely written at least the first time of occurrence. Eg: klebsiella pneumoniae carbapenemase (KPC) and verona integron-encoded metallo-beta-lactamase (VIM)
We thank the referee for this comment. Accordingly, we revised the text.
3. Line 236: Should read as “timely optimization of….”
We thank the referee for this comment. Accordingly, we revised the sentence.
4. Line 240: Should read as “many molecular testing approaches…”
We thank the referee for this comment. Accordingly, we revised the sentence.
5. How does the sensitivity and specificity observed with MALDI-TOF (this study) compare with that of other established tests such as BCID?
We thank the referee for this comment. We compared rapid identification by MALDI-TOF with reference MALDI-TOF identification from pure overnight subcultures. BCID is a molecular system that detect a limited number of bacterial species and KPC gene. Comparing the performance of the rapid MALDI-TOF with that of the molecular test is beyond the scope of this study.
6. Line 327: Italicize E. coli
We thank the referee for this comment. Accordingly, we revised the text.
7. The spelling of “Sepsityper” should be corrected in the figure.
We thank the referee for this comment. Accordingly, we revised the figure.

Round 2
Reviewer 1 Report
All the criticisms raised to the original manuscript were solved by the authors in this revised version.
Reviewer 2 Report
I find the authors' responses to my review satisfactory